# Simulator for Interactive and Effective Organization of Things in Edge Cluster Computing

**DOI:** 10.3390/s21082616

**Published:** 2021-04-08

**Authors:** Woojae Kim, Inbum Jung

**Affiliations:** Department of Computer Information and Communication Engineering, Kangwon National University, Chuncheon 24341, Gangwondo, Korea; wjkimm@kangwon.ac.kr

**Keywords:** edge computing, simulator, virtual things, cluster, resource usage, RaSim

## Abstract

Edge computing is intended to process events that occur at the endpoint of the Internet of Things (IoT) network quickly and intelligently. Edge regions must be organized effectively to facilitate cooperation so that the intention of edge computing can be realized. However, inevitably, many human and material resources are required in the process of arranging things in the edge area to confirm the appropriateness of the thing operation. To address this problem, we proposed a simulator that created a virtual space for edge computing and provided an interactive role and effective organization for edge things. The proposed simulator was aimed at Raspberry Pi as the physical hardware target. To prove the accuracy of the proposed simulator, the similarity between the proposed simulator and the physical target Raspberry Pi was evaluated based on three metrics while executing several applications. In the experiment, several edge-computing service applications were performed in various cluster architecture types formed by the proposed simulator. To support effective resource usage and fast real-time response for edge computing, the proposed simulator identified a suitable number of things in forming the edge cluster.

## 1. Introduction

Cloud computing provides users with a single, logical computing space, enabling users to store, retrieve, and process large amounts of data without restrictions on computer platforms. In particular, because cloud computing provides a specific hardware-platform-independent environment for information processing, it can process a large amount of data generated from the Internet of Things (IoT), in which numerous things are interconnected and operated. However, recent IoT services require intelligent real-time responses to events at the endpoint station of the IoT. To satisfy these new requirements, the storage and processing performance of cloud computing have been improved, but not the network performance. If many people simultaneously access a cloud-computing-based IoT service, access restrictions and delays occur. In IoT services that require intelligent real-time responses, the delay in response time cannot be avoided, and therefore, service quality cannot be satisfied [1,2].

Recently, the concept of edge computing has been proposed to solve the problems that occur in cloud-computing-based IoT services. Edge computing implies that things located at the endpoint of the IoT network directly process and store data. Edge objects are located at the endpoint of the IoT network, and the things process data directly without transmitting them to the cloud or through collaboration with surrounding objects in the edge area, thus solving the problem of processing delay in cloud computing [3,4].

To provide an edge-computing-based IoT service, several things must be arranged, and the arranged things must be horizontally or vertically connected to each other to handle events occurring in a service space. Numerous trial-and-error attempts are required to satisfy these requirements. In addition, testing the collaborative structure of things and providing smooth services is expensive because it requires considerable human and physical resources. Moreover, if the installed devices do not operate as intended, it is expensive to correct them. The cost of additional maintenance and relocation of things is incurred after the service system is deployed. This is also a burden on IoT service operations. To solve these problems, a simulation approach in a virtual space before installing real smart things in a physical IoT environment is required [5].

In this study, a simulator named “RaSim” (Raspberry Pi Simulator) was developed. This simulator can create virtual objects in an edge-computing environment and test their interactions in the virtual space. The proposed simulator is based on a development board called Raspberry Pi 3B+ [6]. Raspberry Pi is widely used in IoT. In particular, Raspberry Pi is equipped with a gigabyte of memory and a quad-core central processing unit (CPU); thus, it has better hardware specifications than existing devices in the IoT environment. Because it is open source, Raspberry Pi has a great advantage in the sharing of development information as numerous libraries and developers work together on joint projects.

In this study, the similarity between the proposed simulator and the physical target Raspberry Pi was evaluated by comparing the battery, bandwidth, and CPU usages. In addition, a comprehensive comparative evaluation was performed using several metrics simultaneously.

To provide various IoT services in an edge-computing environment, several things located in an edge area form a cluster and collaborate through the interactions of the things. Because these edge things are organized into clusters, it is possible to improve the real-time and smart responsiveness of service applications and enhance the longevity of IoT networks. In experiments, according to various metrics, the proposed simulator was used to test the appropriate clustering architectures in edge computing. In addition, methods for effective cluster organization for edge computing were suggested for executing several service applications. These experiments demonstrated that edge clusters can be configured for a variety of service applications in the proposed simulator. The three types of application programs are suitable for evaluating the requirements of edge computing, such as real-time, intelligence, and data processing power.

The remainder of this paper proceeds as follows. In Section 2, edge computing related to this study is explained, the longevity of edges is discussed, and existing simulators are investigated. In Section 3, the proposed RaSim simulator is described. The simulator implementation is described in Section 4. Section 5 describes the performance similarity between the RaSim and Raspberry Pi. In Section 6, edge cluster computing is evaluated by running the service applications on the simulator. Finally, in Section 7, the conclusions and future work are presented.

## 2. Related Work

### 2.1. Edge Computing

Edge computing is a concept used to solve the network bandwidth problem of existing cloud-computing-based IoT environments. Rather than sending data to the cloud and returning the processing result, terminal things are located at the endpoint of the IoT network, processing events occurring in nearby areas directly or through collaboration. As a result, the load on the cloud can be reduced, and the IoT network can guarantee real-time performance.

Cloud computing is composed of high-performance, large-capacity storage to provide various types of services to users, and it enables the processing, storage, management, and analysis of a huge amount of data from the IoT. However, although there have been significant advances in the processing performance and storage space of cloud computing, the network bandwidth problem has not been solved. Therefore, edge computing for urgent events in IoT local areas is a more suitable solution [7,8,9].

Figure 1 shows an edge-computing architecture. Requests and events occurring at the endpoint are processed in the edge cluster rather than being delivered to the cloud data center. Therefore, because the cloud data center does not have to process a large amount of data generated from the endpoint things, network bandwidth usage is reduced. Because the events are processed in the edge cluster, the users in the edge area can receive processing results in real time. An example of the application of edge computing is autonomous driving. In the case of autonomous vehicles, it is necessary to grasp the changing road and surrounding conditions as quickly as possible and to respond quickly through rapid processing. In an edge environment in which cars are driven, safe autonomous driving services are possible only when information on the surrounding environment returns in real time.

Meanwhile, with the advent of such devices as Raspberry Pi and Arduino, which are more economical than the existing devices of things and have relatively high performance, the IoT based on edge computing is laying the foundation for providing new information services to humans. However, even if the devices of things have better performance than before, it is not sufficient for independently processing both the data produced by itself and the data requested from neighboring things. Therefore, there is a need for a clustering method to efficiently process data through the collaboration of things constituting the edges [10,11].

### 2.2. Edge Longevity

In edge computing, the effective usage of battery energy is an important requirement to sustain the longevity of IoT networks. Because things operate on batteries, limited battery power must be used as efficiently as possible. For this purpose, methods such as sensing cycles, unnecessary work minimization, and collaboration can be applied in edge computing. In the case of edge things that include a sensor, power is consumed for event processing and to operate the sensor. Saving the power consumed by operating a sensor can contribute to extending the longevity of the edge network. Sensing power consumption can be adjusted by using a fixed period or sensing by request method. Recently, machine-learning algorithms have been proposed for setting the optimal sensing period based on the available battery power [12].

Since minimizing unnecessary work reduces basic power consumption, it can also contribute to edge longevity. In particular, camera sensors and servo motors consume a considerable amount of power. Power consumption is reduced significantly if these devices are activated by events rather than continuously operating. This reduction in power consumption can contribute to extending edge longevity [13]. In addition, as the edge cluster architecture is introduced, events occurring in the edge area can be assigned to multiple objects in the edge cluster. This approach can extend the longevity of the entire IoT network because the usage of batteries is also distributed in the objects of a cluster. However, the role of things in a cluster structure must be dynamically changed to support a quick response to various events [14].

### 2.3. IoT Simulators

There are three categories of IoT simulators according to the application layer level: A full-stack simulator capable of supporting various elements of the IoT, a simulator focusing on big-data processing, and a network simulator [15]. In the case of network simulators, most were not developed for the IoT but rather for specific scenarios. Subsequently, some functions for the IoT were added. Therefore, in this study, the existing simulators for full-stack and big-data processing are described in related works.

#### 2.3.1. Full-Stack Simulator

The Device Profile for Web Services (DPWS) supports edge devices by providing service description, search, communication, and connection through hosting and hosting services on the Web. DPWSim [16] is a simulator that supports the IoT application development. It is composed of space management, device operation, and event handling functions, and can also test the interaction procedure in DPWS. However, DPWSim is based on the IoT standards, so it is not possible to test new user-defined protocols or existing protocols and technologies.

iFogSim [17] extended CloudSim [18] and was a simulator for evaluating the management and scheduling policies for edge and cloud resources under various scenarios. iFogSim simulates the latency, energy consumption, network congestion, and operating costs on the edges. It informs users of the simulated results based on performance metrics. As an application model, iFogSim uses the sense–process–actuate model. When a sensor publishes data to an IoT network, a fog device subscribes to the corresponding data and processes. The simulation proceeds by controlling the actuators according to the processing results. However, it is inconvenient for users to define the hardware model on the simulator, and it is not possible to simulate multiple IoT application services.

EdgeCloudSim [19] provides an environment for the performance evaluation of edge-computing systems. It is composed of a core simulation module, load generator module, edge orchestrator module, networking module, and mobility module, which is based on CloudSim [18]. Because EdgeCloudSim uses the concept of modules, it is easy to add new features to existing code. However, EdgeCloudSim is based on a logical edge concept instead of a physical hardware device that is currently widely used in industrial fields. Therefore, it is insufficient in terms of risk prediction during actual implementation, which is the basic goal of simulation testing. In addition, the real-time and longevity requirements for edge computing are not covered.

#### 2.3.2. Big-Data Processing Simulator

IOTSim [20] is a simulator extended from CloudSim, and it can test the big-data processing of IoT applications. It applies the MapReduce framework to simulate big-data processing in a cloud-computing environment. MapReduce divides data into blocks of fixed size and processes them in parallel. In this way, IOTSim outputs information related to the job, such as job type, size, processing start time, execution time, and completion time. However, because IOTSim tests cloud-computing-based IoT environments, it does not provide tools to configure edge-computing-based IoT environments, and it is difficult to simulate distributed event processing.

SimIoT [21] is a telemedicine data analysis simulator extended from InterCloud (SimIC) and can process events that occur in smartphones of multiple users. It can simulate delay times and processing completion times. SimIoT controls the number of clients and can transmit sensor data detected by IoT devices to the cloud to show the processing results. However, SimIoT has no capabilities for resource management of IoT devices, and no modules are provided to enable users to test various scenarios.

## 3. Simulator for Edge Computing (RaSim)

In this study, a simulator was proposed for testing interactive and effective cluster organization in an edge-computing environment. The purpose of the simulator is to identify and respond to possible problems before actually doing anything. The proposed simulator’s design is based on Raspberry Pi, which is a widely used physical edge hardware for edge-computing systems. Therefore, it was named as “RaSim” (Raspberry Pi Simulator).

Because RaSim has a simulation role for edge-computing systems, it must be able to simulate algorithms or interactions between virtual things. In addition, it must be able to evaluate the effectiveness of the designed cluster structures. Therefore, the following functions must be provided in the simulator:-It must have functions such as creating, managing, and initializing virtual things;-There must be a function to communicate with virtual things;-It should be possible to determine the resource consumption of each virtual thing;-It must be able to designate the locations of virtual things;-Virtual things must be able to connect with each other;-Connected virtual things must be able to remember each other;-The connection structure of virtual things must be able to change dynamically.

### 3.1. Architecture

Figure 2 shows the architecture of RaSim as a layered model. The layered pattern is a structure in which each layer independently processes tasks and produces results. When developing a system, this layered model pattern has the advantage of shortening the development time and making maintenance easier by focusing on module development for each function. RaSim is divided into three layers: A physical layer, an interaction layer, and a system layer.

**Physical Layer**: As the lowest layer of the simulator architecture, the physical layer is in charge of physical information about things, and it is composed of a thing management module, a sensor management module, and a location module. The thing management module is responsible for creating, initializing, and managing virtual things, and users can check the state of virtual things or test their operations through the application programming interface (API) provided by this module. In addition, it provides information requested by the upper layer. For a thing with physical sensors, the sensor management module manages the sensor type, operation period setting, power consumption, etc. When a data-sensing request is received, the operation is performed on the corresponding modules. Whenever a sensor is added to a virtual thing, the resource usage can be measured according to the operation of the sensors. This module is also responsible for storing and managing the collected sensing data. The proposed simulator models the Raspberry Pi as a physical terminal device for edge computing. This device can simultaneously deploy and control several types of sensors simultaneously. Owing to the various sensors, devices referred to as things can perform various roles in real edge computing. Therefore, virtual things created in the simulation must be able to perform equivalent functions accordingly.

The location module is responsible for the location information of a thing. It can designate or change the location of a virtual thing created through the thing management module. When the mobility of things is tested, it can be implemented through the location module. All location data are stored in the data structure of the virtual thing. This module manages the distance between virtual things and provides the distance information to the higher layers.

**Interaction Layer**: The interaction layer is responsible for the interaction between virtual things. It is composed of a message management module, a link management module, and a discovery module. When a thing requests or receives specific data or tasks toward the neighboring things, the message management module is used. When a message delivery request arrives, it is stored in the message queue. Both the size of the message and the information of the thing to be delivered are analyzed. Subsequently, the message is delivered to the corresponding thing, and the delivered message is deleted from the message queue. Methods for sending messages are divided into unicast, multicast, and broadcast messages. Message types are divided into requests and responses, and specific options such as read, write, and sensor value can be added.

The link management module is responsible for connecting virtual things. The connection information is stored in a data structure to manage virtual things. Based on this information, messages can be delivered to specific things. When the connection is broken or the connection structure is changed, additions and deletions proceed through this module. The thing number can be used to search for other connected things and the name of the sensor can be used to find things that contain a specific sensor. The API provided by the link management module makes it easy to implement complex connections.

The discovery module detects and provides information about the existence of other virtual things around a specific thing within a preset range. Based on this information, the necessary information for the connections between things can be obtained.

**System Layer**: The system layer is in charge of practical operation in the simulation. It is composed of a scheduler module, event module, and resource management module. The scheduler module changes the state of the operation and controls it during the operation. The thing chosen by the scheduler module can read the received message and perform the given task. RaSim is based on a pthread package. Thus, the scheduling method is provided with a round robin and first-come first-serve (FCFS) scheduler. It works with round robin and provides an API to support user-defined scheduling.

The event module manages the events generated during simulation. Events can be defined by the user, such as failure, recovery, and load generation. Defined events can occur for all things or specific things. The event is managed by the event manager, and the user can configure the user-defined event manager to apply the detailed situation. If a user defines an event manager, the default event manager provided by the simulator will not work.

The resource management module calculates the battery usage, Bluetooth bandwidth usage, and CPU usage of each virtual thing from creation to destruction. In the case of battery consumption, the accumulated number of batteries consumed is calculated according to a pre-setting method. Otherwise, this module periodically applies a method to reduce certain battery capacity. In the simulation, the total battery usage includes the standby state, message reception, transmission, and sensing in things.

### 3.2. Data Structure of a Thing

According to the characteristics of IoT service applications, not only does the edge-computing system adjust the number of things that participate in the services, but it also controls the interaction between things. Therefore, a data structure that can express and manage the characteristics of things, connections between things, identification information, and the current state is required.

Table 1 shows a thing control block (TCB), which presents a data structure for each object. Each TCB is created in the form of a thread and consists of three types of information.

**Thing Information**: Thing information contains basic identification information about a thing—information necessary for control, message, and role information. The type of identification information consists of a thread number, thing number, and header number. The thread number is used to obtain the thing information in the simulation, and the thing number and leader number are used to represent the interaction between things. Control information represents the state of a thing and is used for scheduling. The state of things is divided into ready, running, destroyed, and error states. The event module controls the error state, and the scheduler manages the remaining states. Message information is the information of transmitting/receiving data when interacting with other things. The message information includes the sender number, receiver number, transmission method, message type, message path, and sensing value. Role information indicates how things should behave. Currently, the roles of things are classified as init, leader, follower, and spanner. Users can add more roles if they are needed. The basic role of a thing is init, but when connected to another thing, it is classified as either a leader, follower, or spanner. Data processing can vary depending on the role, so the user can specify how the role is handled as desired.

**Physical Information**: Physical information represents the physical information of each item. It is based on Raspberry Pi and is composed of resource information, wireless communication information, owned sensor information, and location information about the thing. Resource information includes information on battery usage, CPU usage, and bandwidth usage. Battery usage includes sensing, communication, and idling. In the case of wireless communication, Bluetooth information is provided as a basic option from the Raspberry Pi version 3. In the case of a sensor, it expresses whether a thing has a sensor, what kind of sensor it is, and how much the sensing period and power consumption of the sensor are. The sensing data are stored here. The location information represents the location of a thing in the virtual space of the simulator. The API for changing location information is provided through the location module, and the thing mobility can be simulated based on this location information.

**Linked information**: Linked information represents information about interconnected things. It includes the connected thing number, distance to the connected thing, role of the thing, and location information. These things have interactions based on this information. The information about connected things is updated whenever the connection structure is changed. The connection structure can vary depending on the service applications of the user. Information on connected things can be searched, added, or deleted through the API provided by the link management module.

## 4. Simulator Implementation

The proposed simulator RaSim was written in C language in a Linux (Ubuntu 16.04 LTS 64 bit) environment. The test environment was constructed in the development environment and Raspberry Pi OS, the official operating system of the Raspberry Pi. In the simulator, each virtual thing is implemented through pthread [22] and can be individually controlled, and the operation process and result can be checked. All sources of RaSim are open (http://snslab.kangwon.ac.kr/v2/RaSim/index.html accessed on 8 April 2021) according to open software regulations [23].

To test the interaction of things through a simulator, there were several stages that had to be performed in common. The first stage was the setup stage of the simulator to meet the conditions of the user. Second, virtual things were created through the creation and initialization stages. Finally, things were connected, and a function was configured to operate according to the role of each thing.

### 4.1. Simulator Setup

Table 2 shows options for the detailed setup of the simulator. The contents of Table 2 are written as constants in the “config.h” and “common.h” header files and can be changed according to the user’s request. In this section, the most frequently used options are described.

MAP_SIZE creates a virtual two-dimensional space to test the interaction of objects in the simulator and is an option for setting the size of the space. The value of the option is expressed as an integer, and the unit is meter. NUMBER_OF_THINGS represents the number of virtual things created in the simulator. BOUNDARY represents the maximum distance when connecting with other surrounding things. IDLE, USING_SENSOR, and USING_BLUETOOTH are options related to battery usage and can be changed and applied according to the measured target. MAX_MESSAGE_SIZE and MAX_Q_SIZE set the maximum size of the message to be sent and maximum size of the message queue, respectively.

In “common.h” options that are generally applied are described, and the state, role, and scheduling method of things can be set.

### 4.2. Virtual Thing Creation and Initialization

Figure 3 shows the pseudocode of the creation and initialization process of the thing after the setup of the simulator was completed. In *initRaSim*, things were created, and all systems were initialized. This process included creating and initializing schedulers and resource managers, as well as allocating memory space referenced by each module. The data input parameters included a location designation method, a reference data path, a scheduling policy, and a user-specified scheduling policy.

*InitTCB* initialized the information of the created thing, the location of the thing, sensor information, and space for the information of connected things. This information could be dynamically changed even during the operation of the simulator.

### 4.3. Role Function

Each thing created in the simulator was assigned a role. Because the role of a thing is just an abstract definition of how the thing operates, a specific operation method had to be specified in the simulator. The reasons for designating the roles of things are as follows. First, the purpose of using the simulator was different for each user. Second, the simulator had to be able to test various methods. Third, in the IoT environment, not all things have a fixed role in their lifetime, and the roles of things may change depending on the current working situation or event. Therefore, flexible coping is needed when providing the role of things. In addition, it is necessary to test the effective operation, interaction, and cooperation of things according to the purpose of the simulator. To this end, role functions had to be configured in the way the user wants, and the intended test had to be completed.

Figure 4 shows the pseudocode for the role function. Each role function was passed to the simulator in the form of a pointer. The role function had a TCB that expresses the thing information as a parameter, and the role function was connected according to the role item specified in the TCB structure. Because the role function pointer had to be applied differently each time the role of a thing changed, the function pointer passed once did not disappear, but it was separately recorded in the TCB and was replaced by a function that fit the role when the role was changed.

When things at the edge formed a cluster, a role was assigned to each object. In this study, the role of things was divided into leaders, followers, and spanners. The leader led the follower and the spanner things included in a cluster. The follower executed the requirements given to the cluster together with the leader. Spanners belonged to several clusters and performed different roles.

Things exist in the init state after creation and are assigned roles in cluster configuration. The user can configure the cluster according to the purpose, and the process of configuring the cluster is user-defined. Therefore, the number of leaders, followers, and spanner varies depending on the algorithm that users want to simulate. Furthermore, the connection structure is not fixed, so the role can change with each change in the connection structure, which is also determined by the user. In other words, each thing can be any of the three roles and can be changed at any time.

### 4.4. Interaction between Things

Figure 5 shows the interactions between the things. During the initialization process, the scheduler changed the state of the thing according to the specified scheduling policy. The changed state checked whether there was a message received from the message queue. If there was a message, the message wsas processed using the currently assigned role function. For example, in Figure 5, in the process of interaction between Things A and C, Thing A, whose status had changed through the scheduler, first checked the message. If it received a message, the message was processed through the role function of Thing A. Otherwise, after requesting specific data from Thing C, it was changed to the standby state. When Thing C changed its state through the scheduler, it checked the message and processed the request of Thing A through the role function. The processed result was delivered to Thing A through the message queue, and the above process was repeated. This process may vary depending on the applied scheduling policy.

Figure 6 shows the message structure used for the interaction between things in the simulator. The thing id is the number of the currently running thing. The sender id is the number of the thing sending the message, and the receiver id is the number of the thing receiving the message. Send type specifies whether to send only to specific things or all connected things. The message type distinguishes whether the message is a request or a response. According to each classification, detailed items, such as sensor value, resource information, and location information, can be added. The sensor value was used when the sensing data were requested. If the data item cannot be expressed as a number, it is transmitted through the message field. Length indicates the total size of the message to be transmitted, and message stores the contents of specific requests or responses.

### 4.5. Simulator Execution and Termination

Figure 7 shows the basic pseudocode for the running simulator. In Figure 7, specific code was omitted for the connection between things and the role of things. The generated virtual things operated in an idle state and calculated the battery usage through the resource manager. The simulation results were returned at the end of the simulation according to the preset options. The results included each virtual thing number and role, battery usage, sensor name, number of sensing, number of message transfers, bandwidth usage, CPU usage, location, and current state. It also printed the sum of the number of messages sent by all things and the number of transmissions for a role. Through this process, we could test the composition, interaction, and resource usage of edge-computing systems.

## 5. Similarity

One reason for using a simulator is that risk management can be performed by pre-testing and evaluating technical problems and expected costs that occur during actual implementation. To meet this purpose as much as possible, the performance evaluation of the proposed simulator should be conducted with hardware and physical conditions like the actual working environment.

This study aimed to simulate an edge-computing system using Raspberry Pi as a standard hardware platform. To evaluate the operation accuracy of the RaSim simulator, a similarity evaluation was performed by comparing the results obtained from RaSim with those obtained with the actual Raspberry Pi. CPU usage, network bandwidth usage, and battery usage were selected as similarity evaluation items.

The CPU and network bandwidth usage are related items to efficiently configure an edge-computing system. On the other hand, battery usage was chosen to evaluate the longevity of the edge-computing systems. Although the selected measurements of the simulator and the actual hardware device cannot be identical, the proposed simulator was deemed excellent if the resulting measurements were highly similar.

### 5.1. Experimental Environment

The Raspberry Pi used in the experiment is a version of Raspberry Pi 3 Model B+, which is a virtual thing reference model of the simulator. Detailed information about the Raspberry Pi is provided in Table 3 [24].

In this study, the similarity between the RaSim simulator and the actual Raspberry Pi was confirmed through three factors: CPU usage, bandwidth, and battery usage. CPU usage was measured by running an algorithm that performs the same operation on the simulator and Raspberry Pi. Bandwidth was measured by the number of data transmitted per second based on the maximum bandwidth of the Bluetooth 4.2 version built into Raspberry Pi. Battery usage was measured in a state of doing nothing (IDLE) to determine the amount consumed while executing a specific algorithm. It was then compared with the actual Raspberry Pi battery usage. The battery usage of Raspberry Pi varies greatly depending on the use of external devices, such as USB ports, displays, or processes running in the background. Therefore, in this study, the experiment was conducted without the use of any external devices or background processes.

The three similarity factors we chose were the ones that were most considered when serving through things with limited performance. Service providers consider computational capabilities (CPU), battery usage, and bandwidth of things to provide IoT services [25]. Based on this, they provide services that meet the capabilities of things. Therefore, three factors were chosen because these characteristics should be reflected in the simulation tester and could be tested in advance.

### 5.2. CPU Usage

Figure 8 shows the CPU usage of the RaSim and Raspberry Pi. In the CPU usage comparison experiment, 20,000 random values were generated, and the usages of selection sort and bubble sort were measured. Because the measured CPU usage could not be used for direct comparison because of differences in clock speed, etc., it was normalized using Equation (1). In addition, it was converted into average values for easy comparison.
(1)$$x_{new}\;=\;\frac{x\;-\;x_{min}}{x_{max}-\;x_{min}}$$

CPU usage data were measured using the Linux “top” command, and the data were used as initial data. The utility “top” can monitor the operating status of the Linux system in real time.

Here, *x_max_* is the largest value among the measured initial data, *x_min_* is the smallest value, and x is the original data. Each of the measured data was converted to *x_new_* using Equation (1) and compared using the average of the converted values. Figure 8 shows that RaSim used approximately 71% of the CPU for selection sort and approximately 62% for bubble sorting. In the case of Raspberry Pi, the selection sort was approximately 60%, and the bubble sort was approximately 53%, with differences of 11% and 9% for selection sort and bubble sort, respectively, compared with the simulator. Comparing the CPU usage of the platforms reveals that the usage of the simulator is 62/71 = 0.87, and that of Raspberry Pi is 53/60 = 0.88. In terms of standard deviation, RaSim has 4.49 and 4.07 standard deviations for selection and bubble sorting. On the other hand, Raspberry Pi are 7.9 and 10.8. RaSim has a smaller standard deviation than Raspberry Pi due to simulator overhead. Therefore, for each sorting algorithm, it was measured that the RaSim used a little more CPU.

Normalized CPU usage does not indicate the actual use of the CPU. A new interval of maximum *x_max_* and minimum *x_min_* was set, and the amount of measured value occupying the corresponding interval was calculated. Therefore, a direct comparison is possible through Equation (1), even though the measurement was performed on heterogeneous CPU platforms. However, the CPU usage was slightly higher in RaSim because of the added overhead of the simulator.

### 5.3. Battery Usage

Figure 9 shows the amount of battery used in 10 min when the virtual thing created in the simulator and Raspberry Pi does nothing. The battery usage of the Raspberry Pi was measured using a KCX-017 voltage current meter. In the case of the virtual thing of the simulator, by referring to the Raspberry Pi battery usage benchmark website and the Raspberry Pi official website, Equation (2) was used.
(2)$$B_{Idle}\;\left(mAh\right)\;=\;+\;\frac{T_{IE}\;\times \;current_{Idle}}{h}$$

In general, it is not possible to simply convert a unit of current (mA) to a unit of battery capacity (mAh) unless the used time is known. The battery usage *B_Idle_* (mAh) in the IDLE state of Equation (2) can be obtained by multiplying the time elapsed since the virtual thing was created with the amount of continuously used current during the time period, and dividing by *h*, which is one hour. 

In Figure 9, the battery usage of the simulator is measured using Equation (2). It used approximately 66 mAh in 10 min. The measured battery usage of the Raspberry Pi was also 66 mAh. The results indicate that RaSim used the same amount of current usage as the Raspberry Pi battery usage benchmark on the official website.

### 5.4. Bandwidth Usage

The bandwidth usage was based on 1 Mbps of Bluetooth version 4.2 built into Raspberry Pi 3 Model B+. The bandwidth usage of a virtual thing can be obtained using Equation (3).
(3)$$Bandwidth\;=\;\frac{D_{Total}}{1\;Mbps\;\times \;T_{TE}}$$

In Equation (3), *D_Total_* is the size of the total transmitted data, and *T_TE_* is the time required to transmit all the data. In this study, a comparison was performed using the average bandwidth obtained through Equation (3), and the total data size was 10 MB.

Figure 10 shows the bandwidth usage of the RaSim and Raspberry Pi. It was measured at approximately 41% for RaSim and 51% for Raspberry Pi. Although Bluetooth is a representative wireless communication protocol, the delay time may not be constant, even when data of the same size are transmitted, owing to obstacles or interference from radio waves. Furthermore, the distance between the communicating things affects the delay time. However, it is difficult to predict these items in advance or calculate them each time and apply them to a simulator. In the future, research on the conditions affected by wireless communication should be conducted through experiments in various operating environments.
(4)$$Delay_{Transmission}=\;\frac{D_{Total}}{1\;Mbps}$$

In RaSim, the delay time model of Equation (4) was applied. Equation (4) gives the total data size divided by the standard bandwidth of 1 Mbps. The results in Figure 10 show that the bandwidth usage of RaSim and Raspberry Pi had a difference of approximately 10%. External factors mentioned above, such as radio wave interference, are presumed to be the cause of the bandwidth increase in real Raspberry Pi.

### 5.5. Resource Usage by Interaction between Things

Figure 11 shows the resource usage of each thing when things interact. The comparison items are the CPU usage and battery usage of each item when transferring 10 MB of data. First, in the RaSim shown in Figure 11, the cases of virtual things were divided into sender and receiver. In the case of the sender, the measured CPU usage was approximately 34%, and the battery usage was approximately 12 mAh. For the receiver, the measured CPU usage was approximately 65%, and the battery usage was also approximately 12 mAh.

Raspberry Pi was also divided into the sender and receiver cases. In the case of the sender, the measured CPU usage was approximately 50%, and the battery usage was approximately 18 mAh. The measured CPU usage of the Raspberry Pi receiver was approximately 85%, and the measured battery usage was also approximately 18 mAh. The receiver CPU usage was higher for both RaSim and Raspberry Pi. This is because the method of processing the received data was added here. When the CPU usage is compared in terms of the sender and receiver of RaSim and Raspberry Pi individually, RaSim differs by approximately 31%, and Raspberry Pi differs by approximately 35%. These differences seemed to be quite large numerically; however, the ratio of the sender and receiver was quite close, with values of 34/65 = 0.52 in RaSim and 50/85 = 0.58 in Raspberry Pi. Therefore, the RaSim simulator reflected the characteristics of the Raspberry Pi well because both the ratio and trend of CPU usage in the sender and receiver aspects were quite similar.
(5)$$B_{Total}\left(mAh\right)\;=B_{Idle}\;+\;\frac{Delay_{Transmission}\;\times \;current_{TX}}{h}$$

The battery usage of a virtual thing is measured using Equation (5). Here, *B_Idle_* is the battery usage when the virtual thing is doing nothing, *Delay_Transmission_* is the transmission delay time, *Current_Tx_* is the maximum current used for one transmission, and *h* is one hour. 

As shown in Figure 11, for RaSim, 12 mAh was measured for both the transmitter and receiver. For Raspberry Pi, 18 mAh was measured for both the sender and receiver. The difference of 6 mAh was the result of the accumulated delay time in transmitting data. This means that an overhead occurs in the actual working Raspberry Pi. The overhead is analyzed in terms of time. The Raspberry Pi took 163 s to transmit the data, whereas RaSim transmitted the data in 144 s. Therefore, 19 s was regarded as an overhead in the Raspberry Pi operation. If the amount of battery consumed for the IDLE status and message transmission was added or subtracted, the actual trend of resource usage by Raspberry Pi and RaSim was similar.

## 6. Edge Clustering

### 6.1. Architecture

Currently, the computing power of a single thing used in an edge area is insufficient to satisfy various service requirements at the edge. Therefore, to meet the requirements of edge service applications such as real-time, longevity, it is necessary to connect with other things to form a cluster for collaboration. Figure 12 shows a cluster model formed in an edge-computing environment using RaSim. The physical area in Figure 12 is the cluster configuration when the actual things are deployed. The conceptual area shows that the physical arrangement of the environment is divided into several layers. It represents various edge-computing connection configurations to support several edge service applications. One cluster is in charge of one application. When an application is terminated, the things of the cluster can be relocated to another cluster to run other remaining applications.

In this figure, things are divided into leaders, followers, and spanners. To support various edge service applications simultaneously, the things belonging to one cluster also belonged to other clusters, and their roles could be continuously changed. The leader had information about other things connected to itself by leading other things to provide services to users. A follower was connected to one leader in one cluster and handled the leader’s requirements for service provision. A spanner is a thing belonging to multiple clusters and simultaneously plays the role of linking with other clusters and handling events. In addition, in a specific cluster, the spanner can be a leader simultaneously, as indicated by “*L*/*S*” in Figure 12. Because the spanner may have to perform more operations than the leader in some cases, research on the selection of the spanner was necessary.

In order to form a cluster to handle service demands occurring in the edge area, the characteristics of the things included in the cluster and the surrounding environment had to be considered. In addition, to satisfy the real-time processing and longevity requirements in edge computing, the cluster architecture had to be organized for the characteristics of the service applications, and the role exchanging/sharing and resource utilization between things had to be effectively designed.

In this study, three types of service applications were selected to evaluate the interaction between edge things and effective resource usage in edge cluster computing. The key algorithms of these applications were executed using various edge cluster architecture types. First, we selected a big-data processing application to handle the large amount of sensing data generated in the edge area. The key algorithms of this application were sorting and searching operations. Second, we selected a machine-learning application based on a backpropagation neural network algorithm. This application provided an intelligent response to events occurring in the edge area. Third, a diagnostic application was selected. The key diagnostic algorithm was the fast Fourier transform (FFT) program, which was used in various types of diagnostic service applications such as magnetic resonance imaging (MRI), computed tomography (CT), communication, non-destructive testing equipment, aviation radar, physical exploration, and structure monitoring.

In the simulation, the cluster architectures for effective resource usage were evaluated in terms of processing time, battery usage, and throughput. As a result, the most suitable cluster size for a given service application was suggested. In addition, we evaluated the real-time and longevity properties, which are the representative requirements of edge computing.

### 6.2. Big-Data Processing Application (Data Sorting/Searching)

In big-data processing applications, the key process is to quickly find the information the user wants from the big-data location. In this experiment, to evaluate the differences in performance according to the cluster architecture types, bubble-sorting and binary searching were performed to find the information desired by the user.

Table 4 lists the experimental parameters of the big-data processing application. It distributed 500,000 unsorted data to each cluster and performed sorting and searching operations. The cluster architecture type was set to 5, and the number of items included in each cluster type were 16, 32, 64, 128, and 256 for clusters 1, 2, 3, 4, and 5, respectively. The total task number was set to 50, and each cluster was initially allocated a task. However, clusters that had finished the initial task performed the remaining tasks according to the FCFS policy.

#### 6.2.1. Processing Time

Edge computing uses a cluster architecture to support fast real-time processing of events occurring in the edge area. Figure 13 shows the average processing time measured by cluster type. Among the clusters, cluster 1 showed the longest average processing time of 58.5 s. Cluster 3 exhibited the shortest average processing time, with an average processing time of 15.9 s. As shown in Table 4, the number of things used in cluster 3 was 64. In cluster 4 and cluster 5, more things participated in cluster architecture than cluster 3, but the average processing time was longer than in cluster 3. As the number of things participating in the cluster increased, the information exchange required for the distributed processing of tasks between things gradually increased, and as a result, the total average processing time increased.

To evaluate the real-time responsiveness in edge cluster computing, we performed the same big-data processing load on a single thing, and the average processing time was 803 s. As shown in Figure 13, the average processing time in clusters was much faster than 803 s for a single thing; thus, we confirmed that the cluster architecture can contribute to the real-time processing requirements of edge computing.

In the experiment results, cluster 3, composed of 64 things, provided the fastest real-time response. However, because the high-performing cluster architecture can be changed according to the characteristics of the application, it is necessary to pre-verify the expected performance in the simulator before implementing the application service using real hardware such as Raspberry Pi.

#### 6.2.2. Battery Usage

To maintain the longevity of edge computing as much as possible, it is necessary to measure and evaluate the battery usage of things. Figure 14 shows the average battery usage in each cluster after task execution. When the same task load was performed on a single thing, the average battery usage was 2258 mAh. Figure 14 shows that more battery usage is required in cluster structures than in a single thing. This is because the things forming the cluster jointly process tasks, thereby increasing data communication for mutual information exchange. Therefore, battery usage increases.

In Figure 14, among the cluster types, cluster 3 had the least average battery usage and cluster 5 had the most average battery usage. However, the difference in battery usage between cluster types was not significant. Based on the results shown in Figure 13 and Figure 14, considering both the real-time and longevity of edge computing, the most suitable edge cluster size in the current simulation environment was cluster 3, which was composed of 64 things.

#### 6.2.3. Throughput

Figure 15 shows the throughput of the tasks in each cluster structure. It shows the highest throughput was in cluster 3, which had the shortest average processing time (as seen in Figure 13). It shows that 14 of the 50 tasks were processed in cluster 3.

### 6.3. Machine Learning Application (Backpropagation Neural Network)

A fast response to events occurring in the edge area is also required, and edge service applications that require smart decisions in the edge area are also increasing. In this experiment, a backpropagation (BP) neural network algorithm was selected among several machine-learning algorithms used for smart information processing, and its performance was evaluated in the edge clusters. Table 5 lists the experimental parameters of the BP neural network algorithm. The learning data were information on the sensing period and battery usage of things deployed in the edge region and could be generated as described in [12]. Using learning data, this algorithm learned the sensing period that optimized battery consumption in each edge-computing environment.

The number of learning data used was 50,000, and each cluster was learned by distributing the data. In the neural network structure, 10 nodes constituted one hidden layer, and the activation function of the hidden layer used ReLU. The output layer consisted of one node, and the activation function used a sigmoid.

An advantage of neural network learning methods is that neural network configurations can be set differently depending on hardware performance. Raspberry Pi, the physical target of virtual things, has limited performance. Therefore, a small neural network was constructed [12,26].

#### 6.3.1. Processing Time

Figure 16 shows the average processing time required to complete the learning in edge clusters. Cluster 1 showed a processing time of approximately 337 s, which is the longest learning time. The shortest processing time was that of cluster 5, which was approximately 38 s. Figure 16 demonstrates that as the size of the cluster increases, the learning completes faster. Conversely, when learning on a single thing using the same learning data, a processing time of approximately 652 s was measured. The edge clustering method can learn up to 17 times faster than the learning time in a single thing. This is because as the size of the cluster increases, the size of the training data distributed to the things participating in each cluster decreases, and thus the time required for learning also decreases.

#### 6.3.2. Battery Usage

Figure 17 shows the average battery usage in each cluster when the tasks are completed in clusters. The battery usage of cluster 5 was the highest at 1104 mAh, and the remaining clusters showed similar levels of battery usage. On the other hand, when performing on a single thing, the battery usage was measured at 1068 mAh. In Figure 16, which is the result of the previous experiment, cluster 5 was able to learn 17 times faster than a single thing. However, in the case of battery usage, it was found that cluster 5 requires somewhat more energy than a single thing. This demonstrates that to extend the longevity of edge computing, it is more effective to select smaller cluster types than cluster 5.

#### 6.3.3. Learning Accuracy

Figure 18 shows the accuracy of the model trained on each cluster type. In the case of the edge cluster, an accuracy of 99.67% is achieved in cluster 1, which is the highest value among the clusters. Cluster 5 achieves 98.86% accuracy and has the lowest accuracy among the clusters. As the number of things constituting a cluster increases, the size of the data that each thing needed to learn decreased. If the number of learning data was too small, it was difficult to predict various cases, and thus, the accuracy was degraded. On the other hand, when the learning model was tested on a single thing, it showed a high accuracy of 99.7%, which was higher than that of clusters.

From these experimental results, we demonstrated that in the case of an edge service application using backpropagation neural network machine learning, the cluster size should be set according to the level of requirements in service applications, such as real-time processing, longevity, and learning accuracy.

#### 6.3.4. Throughput

Figure 19 shows the throughput of the clusters. As the cluster size increased, more tasks could be processed. Cluster 1, composed of 4 things, had the smallest size among the clusters and shows the lowest throughput. Cluster 5 was the largest among the clusters, with 64 things participating and obtaining the highest throughput. Therefore, in terms of throughput, cluster 5 was the most effective. However, as shown in Figure 17 and Figure 18, the longevity and accuracy were affected by the size of the cluster, so the size of the cluster had to be adjusted according to the requirements of the edge service application.

### 6.4. Diagnostic Application (FFT)

In diagnostic application programs, an FFT algorithm was used to determine the situation of generated events by converting data measured in the time domain into the frequency domain. In particular, if FFT results were obtained quickly from sensing data—indicating an emergency—an effective response was possible in diagnostic applications. Table 6 lists the experimental parameters of the FFT program used in the experiment. A 1024 × 1024 matrix was used as an input to the FFT. The number of operations in the FFT program varied greatly depending on the data size, and this characteristic affected the real-time and longevity of edge computing.

#### 6.4.1. Processing Time

Figure 20 shows the average processing time when performing tasks on an edge cluster. Cluster 1 had the longest processing time (37.5 s), and cluster 5 had the shortest processing time (2.1 s). When running on a single thing with the same load, a processing time of 101 s was measured. Compared to execution on a single thing, the processing speed of cluster 1 was 2.7 times faster and cluster 5 was 48.1 times faster. For emergency medical applications that require emergency services, cluster 5 was a suitable structure that could provide the fastest real-time performance.

#### 6.4.2. Battery Usage

Figure 21 shows the average battery usage in edge clusters. Among the clusters, cluster 1 had the lowest battery usage and cluster 5 had the most. When running on a single thing using the same load, 2633 mAh was measured. From the battery usage measurement results, we found that all cluster types used more batteries than a single thing. In addition, Figure 21 shows that the larger the cluster size, the larger the number of participating things, resulting in a slight increase in battery usage. This is because the battery usage increased when the FFT algorithm ended the calculation of one step and all the things had to synchronize to proceed to the next step. However, when evaluating the experimental results, cluster 5 used approximately 1% more battery than cluster 1. In other words, there was little difference in battery usage between cluster types.

In terms of edge-computing longevity, the cluster type did not have a significant impact on this requirement when the diagnostic applications, including the FFT algorithm, were running. Therefore, in emergency diagnostic applications using the FFT algorithm, it was effective to determine the edge cluster size based on processing time rather than battery usage.

#### 6.4.3. Throughput

Figure 22 shows the throughput of the clusters. In the FFT algorithm, the number of operations varied greatly depending on the data size, so the throughput increased as the cluster type increased. As shown in Figure 22, cluster 5 was able to process 24 out of 50 tasks.

Owing to the characteristics of the FFT algorithm, the larger the cluster size, the higher the throughput. Considering the real-time and longevity of edge computing based on the results of Figure 20 and Figure 21, cluster 5 was the most suitable structure under current working conditions.

## 7. Conclusions and Future Work

In this paper, a simulator called RaSim was proposed to support effective cluster organization of things for edge cluster computing. RaSim can simulate the Raspberry Pi as a real hardware platform model, which is usually adopted to implement IoT edge service applications. Through experiments, the similarity between the simulator RaSim and the real Raspberry Pi was evaluated, and the real-time, longevity, and throughput requirements were tested on various types of edge cluster architectures.

For similarity evaluation, CPU usage, battery usage, bandwidth usage, and resource usage by interaction between things were measured for both RaSim and Raspberry Pi. CPU usage was measured using selection-sorting and bubble-sorting with 20,000 data points. It was determined that there was a difference of 11% for selection sorting and 9% for bubble sorting between RaSim and Raspberry Pi. There was no difference in battery usage, as both RaSim and Raspberry measured battery usage was 66 mAh in the IDLE state for 10 min. The bandwidth usage was measured while transmitting and receiving 10 MB of data, and it was found to be 41% for RaSim and 51% for Raspberry Pi. In both RaSim and Raspberry Pi, the receiver CPU usage was higher, and the usage trends of the sender were quite similar. In the similarity experiment, depending on the workload, the resource usage measured in RaSim and Raspberry Pi both increased and decreased in the same trend. In addition, there was only a small difference between increasing and decreasing resource usage. Similarity evaluation results showed that RaSim can serve as a simulator for edge-computing systems.

Based on the similarity evaluation, edge clusters were constructed using the RaSim simulator, and performance evaluation was conducted by selecting three types of applications. The selected applications were big-data processing applications, machine-learning applications, and diagnostic applications. The key algorithms of these applications were executed in various edge cluster architecture types to measure the processing time, battery usage, and throughput. In particular, in machine-learning applications, the accuracy of the learning results was added as a metric for performance evaluation. By executing the applications, it was confirmed that the real-time and longevity requirements of edge computing can be improved by finding suitable edge cluster types to reflect the characteristics of the applications. Before launching an edge service application directly on-site, key algorithms can be pre-evaluated using the RaSim simulator. Through this role, RaSim can contribute to the selection of a suitable cluster architecture for various edge service applications, as well as verifying the expected performance and effective resource use in the edge cluster.

Because the things in edge computing are placed at the endpoint of the IoT, they suffer from unstable operating environments, such as wireless communication interference, uncertainty in energy supply, unbalanced work distribution, restoration of dead things, and maintenance of changed software consistency. These problems cause a variety of requirements, so it is necessary to analyze them in terms of the architecture drivers of edge computing. Therefore, we intend to proceed with future research to evaluate real-time, longevity, availability, reliability, etc. in advance through the RaSim simulator. Furthermore, to simulate interactions between various things, we will analyze the behavior of things such as Raspberry Pi 4 and Arduino and apply it to the simulator.

## Figures and Tables

**Figure 1 sensors-21-02616-f001:**
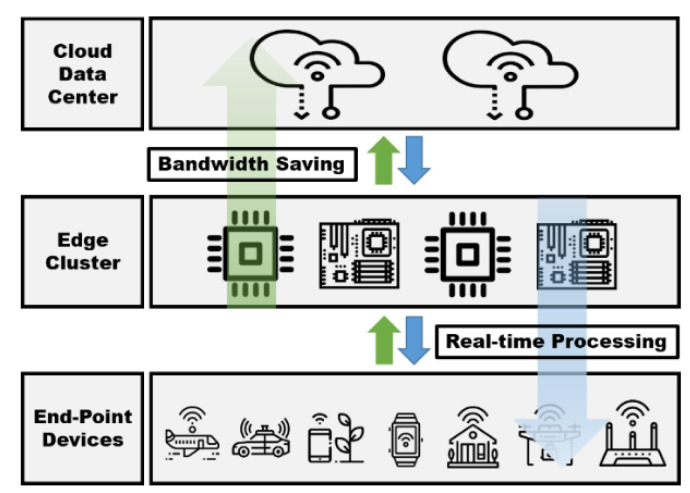
Edge-computing architecture.

**Figure 2 sensors-21-02616-f002:**
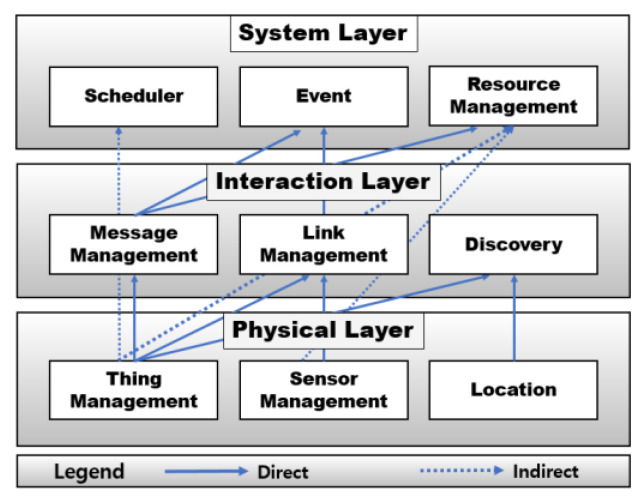
RaSim architecture (layered model).

**Figure 3 sensors-21-02616-f003:**
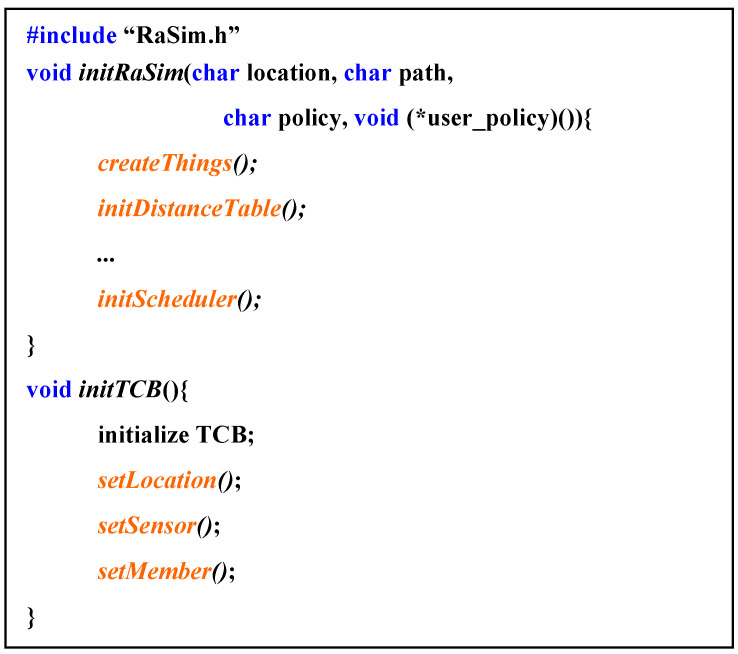
Pseudocode for “create virtual things and initialize”.

**Figure 4 sensors-21-02616-f004:**
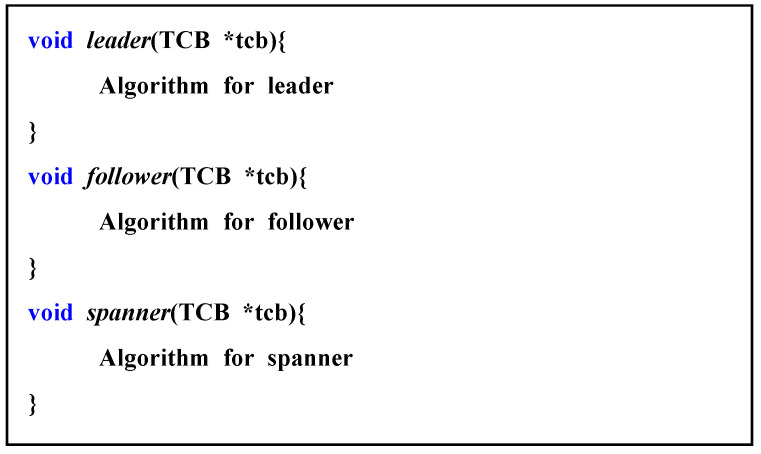
Pseudo code for role function.

**Figure 5 sensors-21-02616-f005:**
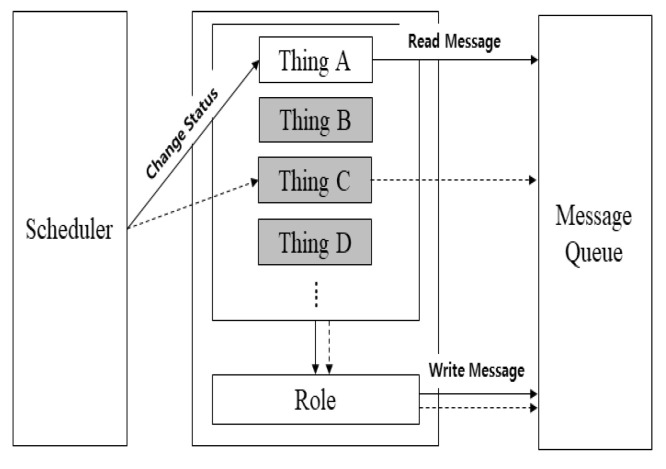
Interaction between things.

**Figure 6 sensors-21-02616-f006:**
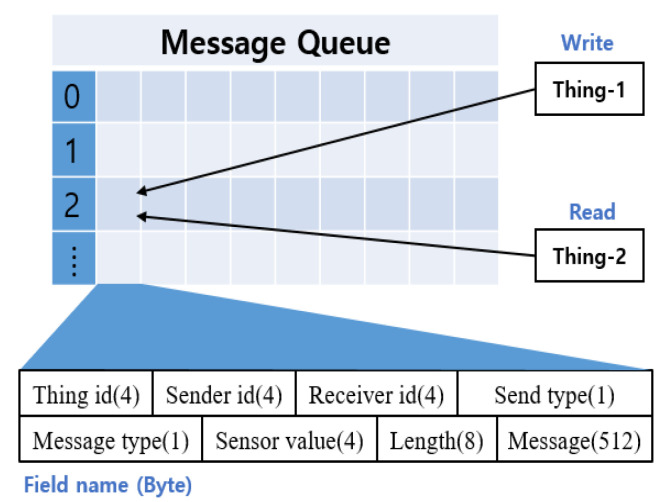
Message structure for interaction.

**Figure 7 sensors-21-02616-f007:**
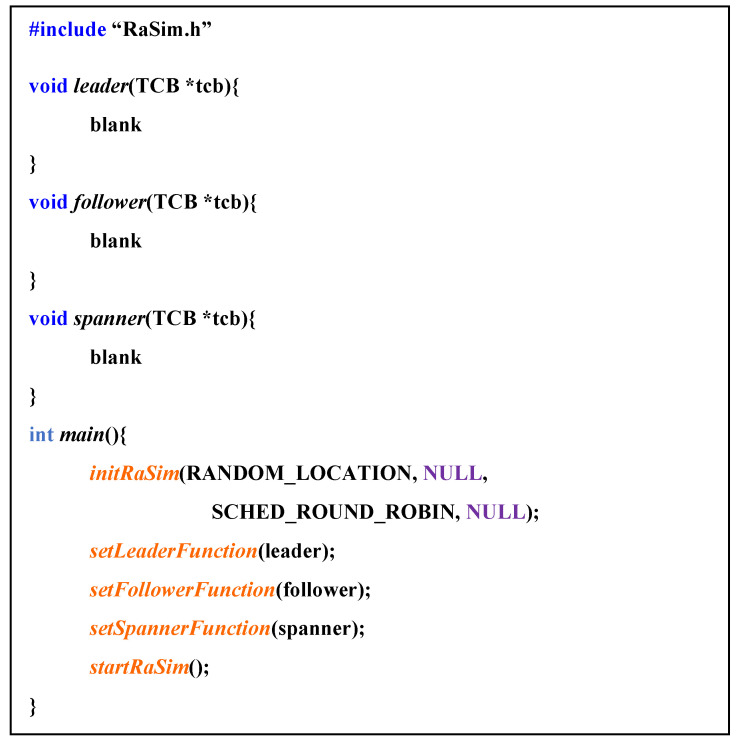
Pseudo code for running simulator.

**Figure 8 sensors-21-02616-f008:**
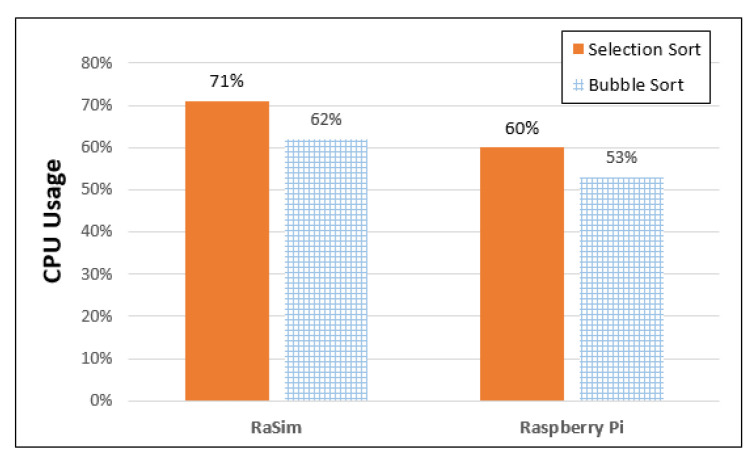
CPU usage of RaSim and Raspberry Pi.

**Figure 9 sensors-21-02616-f009:**
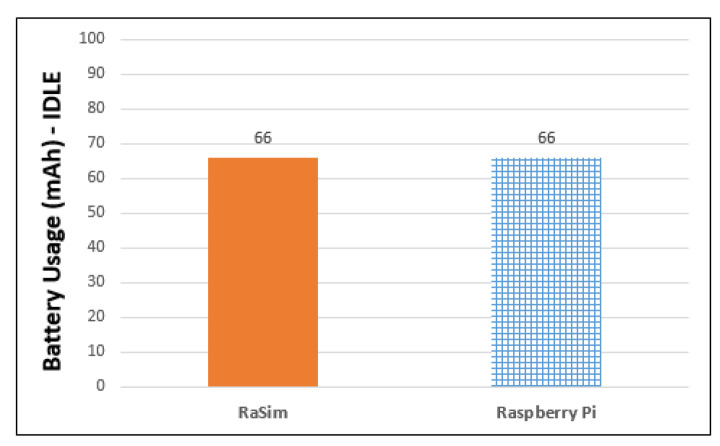
Battery usage of RaSim and Raspberry Pi.

**Figure 10 sensors-21-02616-f010:**
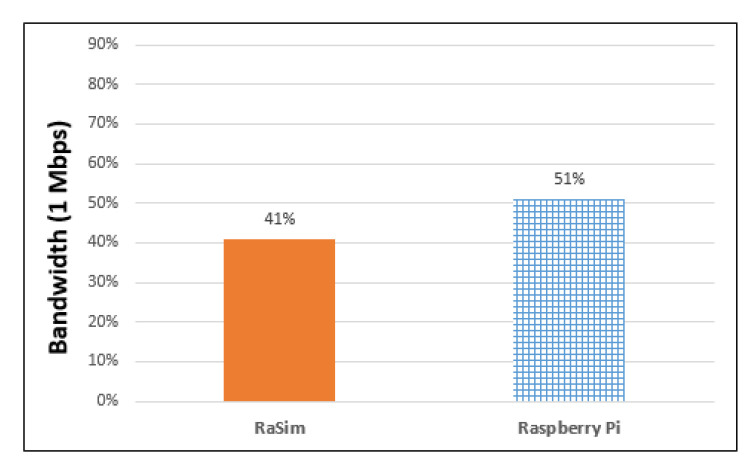
Bandwidth usage of RaSim and Raspberry Pi.

**Figure 11 sensors-21-02616-f011:**
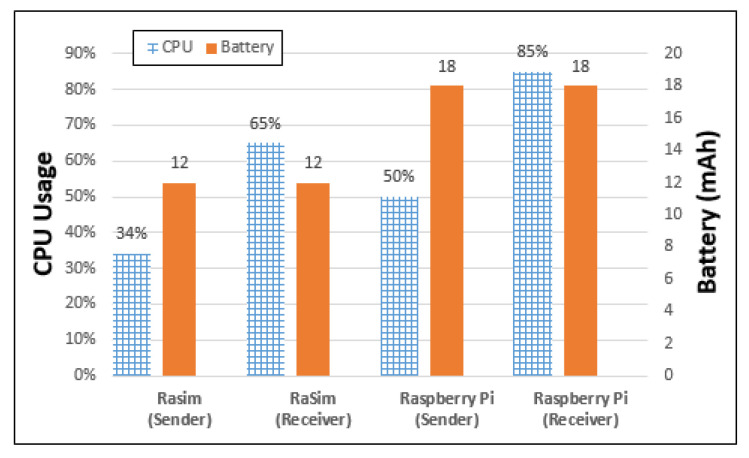
Resource usage by interaction between things.

**Figure 12 sensors-21-02616-f012:**
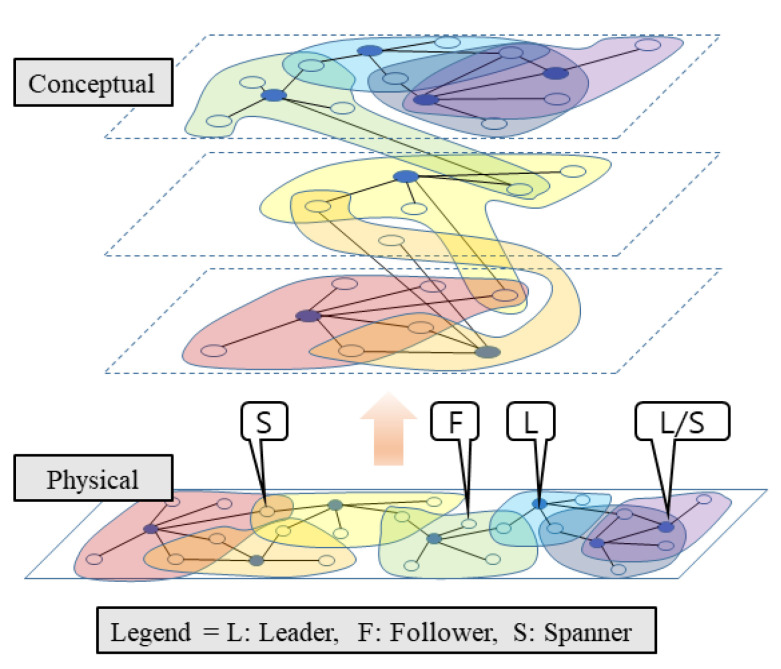
Cluster configuration in edge-computing environment.

**Figure 13 sensors-21-02616-f013:**
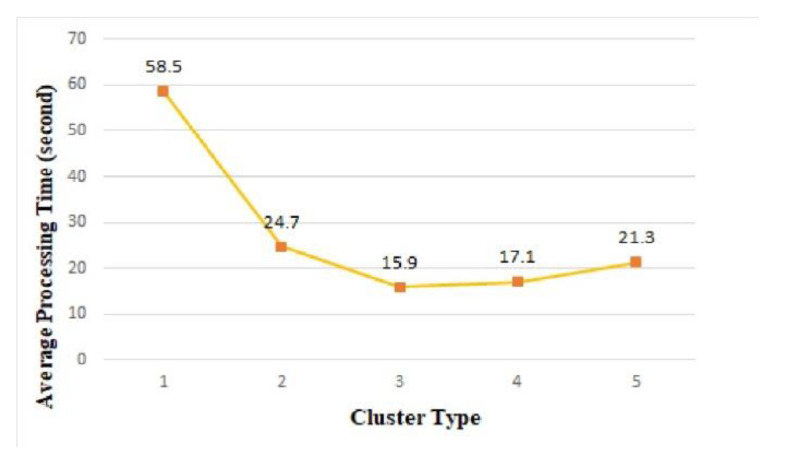
Average processing time of edge clusters.

**Figure 14 sensors-21-02616-f014:**
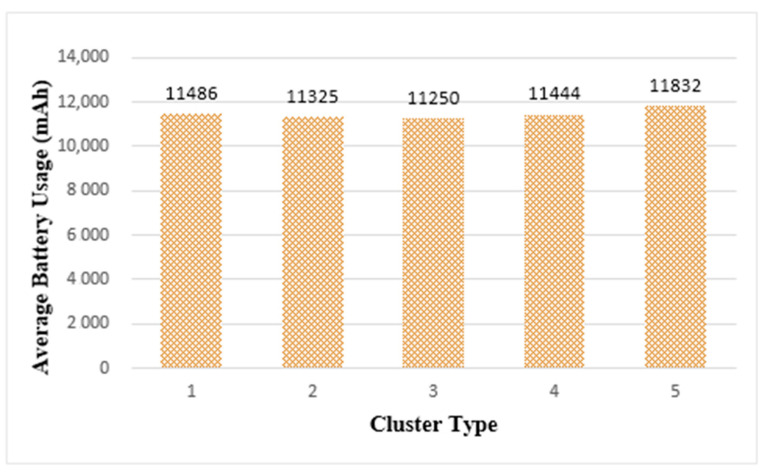
Average Battery Usage of Edge Clusters.

**Figure 15 sensors-21-02616-f015:**
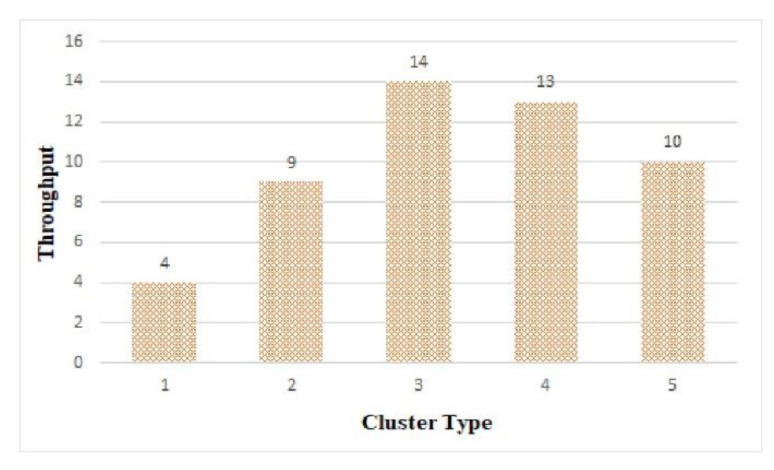
Throughput by cluster.

**Figure 16 sensors-21-02616-f016:**
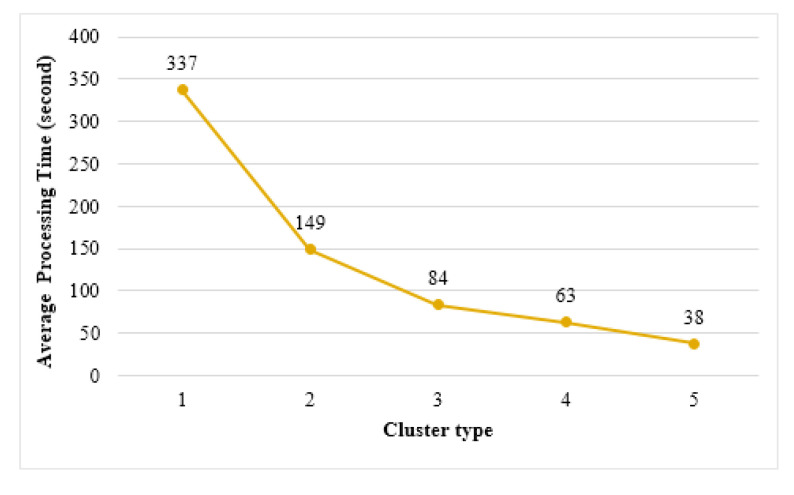
Average learning time of edge clusters.

**Figure 17 sensors-21-02616-f017:**
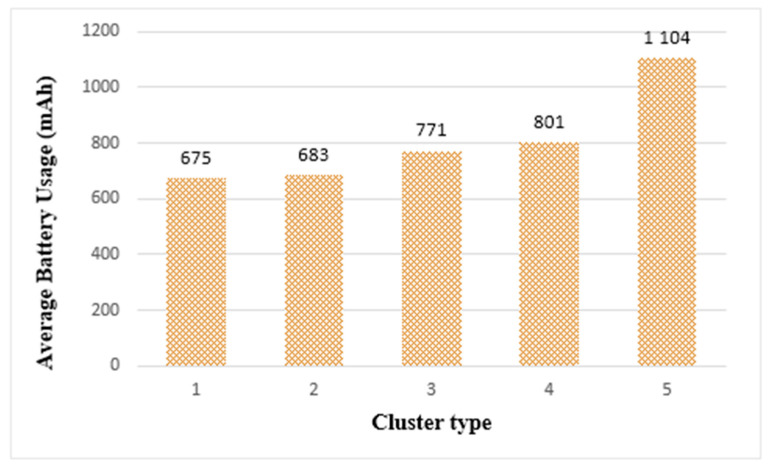
Average battery usage of edge clusters.

**Figure 18 sensors-21-02616-f018:**
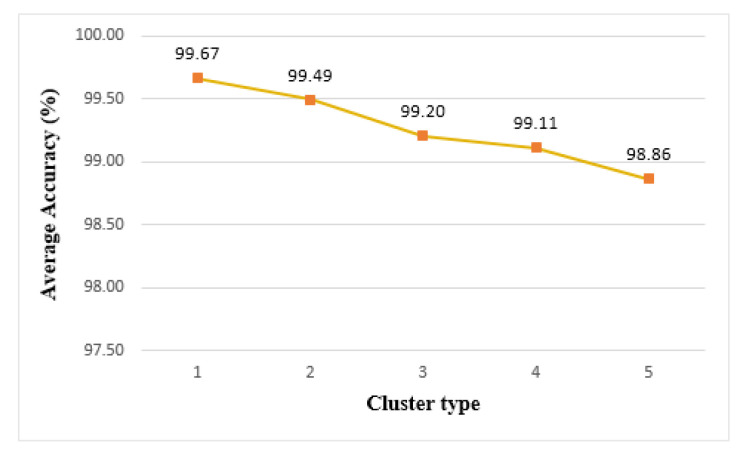
Average accuracy of edge clusters.

**Figure 19 sensors-21-02616-f019:**
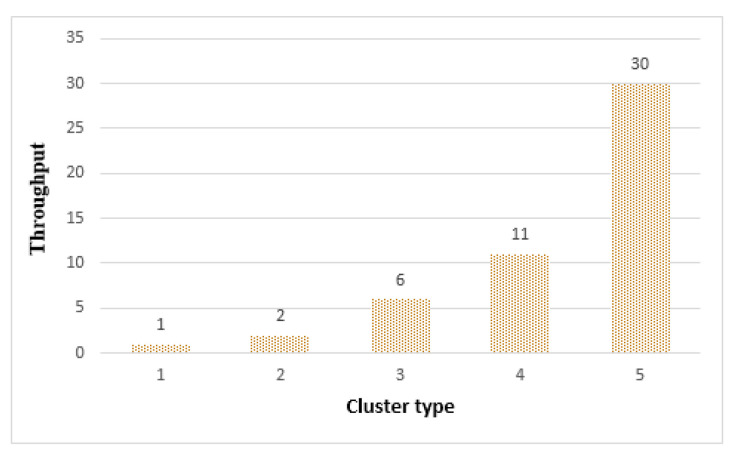
Throughput by cluster.

**Figure 20 sensors-21-02616-f020:**
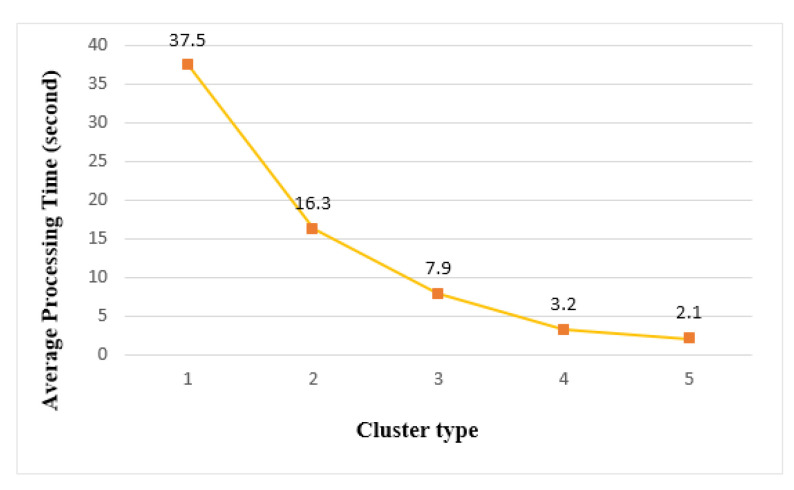
Average processing time of edge clusters.

**Figure 21 sensors-21-02616-f021:**
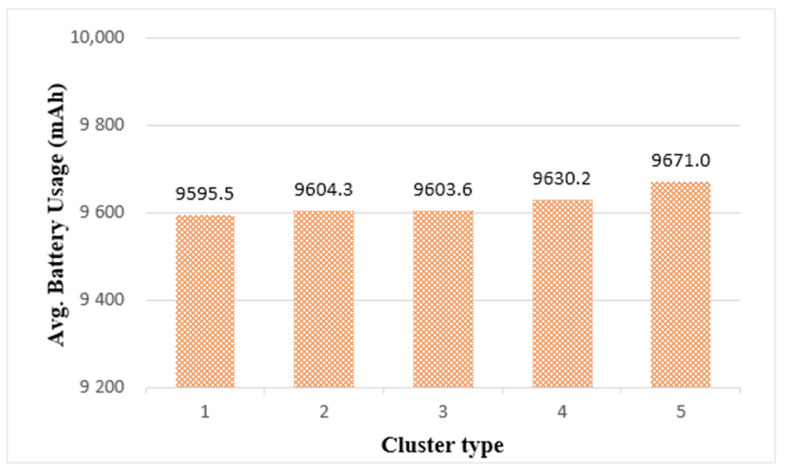
Average battery usage of edge clusters.

**Figure 22 sensors-21-02616-f022:**
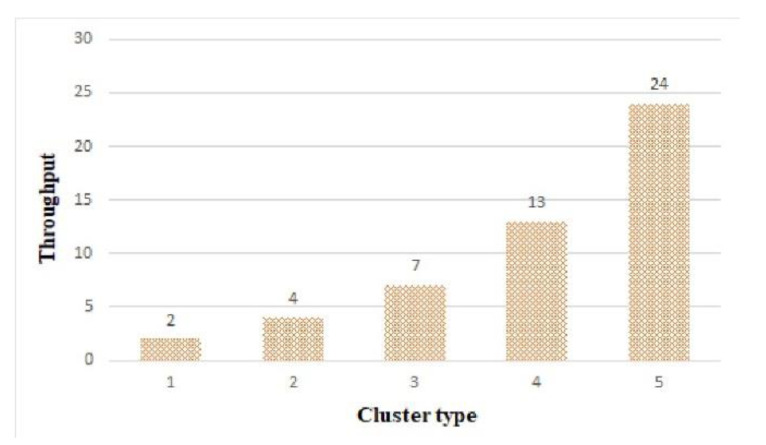
Throughput by cluster.

**Table 1 sensors-21-02616-t001:** Thing control block (TCB) structure.

Thing Infomation	Thread ID	Thing ID
	Leader ID	Status
	Role	
	Message	
	Number of transmissions	
Physical Infomation	Resource (battery, CPU, bandwidth)	
	Bluetooth	
	Sensor	Location
Linked Infomation	Member link	
	Reserve(n)	

**Table 2 sensors-21-02616-t002:** Part of option for RaSim setting.

<config.h>	<common.h>
#define MAP_SIZE	#define DESTROYED
#define NUMBER_OF_THINGS	#define LEADER
#define BOUNDARY	#define FOLLOWER
#define IDLE	#define SPANNER
#define USING_SENSOR	#define SCHED_ROUND_ROBIN
#define USING_BLUETOOTH	#define SCHED_FCFS
#define MAX_MESSAGE_SIZE	#define SCHED_USER_POLICY
#define MAX_Q_SIZE	#define BROADCAST
⁝	⁝

**Table 3 sensors-21-02616-t003:** Raspberry Pi 3 Model B+ specification.

Category	Description
Product	Raspberry Pi 3 Model B+
CPU	Cortex-A53(ARMv8) Quad core @1.4 GHz
RAM	1 GB
Bluetooth	Version 4.2, BLE
Power requirements	5 V/2.5 A DC power input
Power consumption	Idle ≈ 400 mA (Depends on use case)

**Table 4 sensors-21-02616-t004:** Experimental parameters for big-data application.

Parameter	Description
Data size (set)	500,000
Cluster architecture types	5
Number of things by cluster type	16, 32, 64, 128, 256
Number of tasks	50
Time complexity of sorting algorithm	$O\left(N^{2}\right)$–Bubble Sort
Time complexity of searching algorithm	O(logN)–Binary Search

**Table 5 sensors-21-02616-t005:** Experimental parameters for machine learning.

Parameter	Description
Data size (set)	50,000
Cluster architecture types	5
Number of things by cluster type	4, 8, 16, 32, 64
Number of tasks	50
Hidden layer (number of nodes)	1 (10)
Output node	1
Hidden layer activation function	ReLU
Output layer activation function	Sigmoid

**Table 6 sensors-21-02616-t006:** Experimental parameters for diagnostic application.

Parameter	Description
Data size (set)	1024 × 1024
Cluster architecture types	5
Number of things by cluster	4, 8, 16, 32, 64
Number of tasks	50

## Data Availability

Not application.

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
