# Peer review of "Simulator for Interactive and Effective Organization of Things in Edge Cluster Computing"

_sensors, 2021, doi:10.3390/s21082616_

Round 1
Reviewer 1 Report
The authors present the development of a simulator for edge computing. They intend to show how this simulator can correspond to a physical
target. To do so, they describe a similarity between the Simulator and the physical target, a Raspberry Pi.
They also use some metrics to draw a comparison, metrics such as the battery, bandwidth, and CPU usage.
Besides, this work aims to discuss effective cluster organisation, where three roles of the interactive relationship between devices on the edge
are defined. In general, the paper is well presented and seems to bring some contributions.
-----------------------------------------------
# Key Issues:
1. The Related Work section presents some concepts of edge computing and brief analysis of two kinds of simulators. I would suggest improving
this section by describing and comparing related work according to the goal of this work, since as I understood the goal of this work is not
only the simulator but also edge cluster organisation. So, it is expected to find a detailed discussion on all intended contributions of this work.
2. Is it possible to extend the simulator to use it on different target hardware (besides Raspberry)? How hard would it be? This could be discussed in the section of future work.
3. In Section 6, it is said that three types of applications were selected to assess the system. Why these three kinds of applications were chosen? What is intended to present with these applications? These could be clearly stated to the reader in the Introduction.
-----------------------------------------------
As it follows, some specific comments and corrections:
# 1. Introduction
> Suggestion: describe in details the intended contributions of this work. Is the simulator the main contribution? What else?
# 4. Simulator Implementation
> Line 454, Subsection 4.5 has the same font style as a Section name.
> It could be given details about how the algorithm works to set up a leader, follower, and spanner. For example, how many followers exist?
When and how a thing can have its role changed? How this can role can be changed? Can a follower became a leader?
# 7. Conclusions
> Why selection-sorting and bubble-sorting were selected to measure CPU? Is this a typical operation in a cloud environment? Why not use communication and interaction operations for example? Especially to measure bandwidth?
> Line 867 mentions the use of selection-sorting and bubble-sorting algorithms, but in line 686 bubble-sorting and binary search are mentioned.
Which pair of algorithms were selected?
> The future work could be detailed. How exactly the RaSim simulator can be improved?
Reviewer 2 Report
The article proposed a simulator called RaSim to support effective cluster organization of things for edge cluster computing. The simulator creates a virtual space for edge computing and provides an interactive role and effective organization for edge things. RaSim can simulate the Raspberry Pi as a real hardware platform model, which is usually adopted to implement IoT edge service applications. To prove the accuracy of the proposed simulator, the similarity between the proposed simulator and the physical target Raspberry Pi is evaluated based on three metrics while executing several applications. From a relative viewpoint, the experiment shows that the central processing unit, battery, and bandwidth usage of the virtual objects are like the physical object measurements in the Raspberry Pi. Based on the similarity evaluation, edge cluster computing provides effective resource usage and interactive operations among things when organizing edge clusters. The concepts of leader, spinner, and follower are introduced to assign the roles of individual things in a cluster architecture.
In the experiment, several edge-computing service applications are performed in various cluster architecture types formed by the proposed simulator. The selected applications were big-data processing applications, machine-learning applications, and diagnostic applications. In machine-learning applications, the accuracy of the learning results was added as a metric for performance evaluation. By executing the applications, it was confirmed that the real-time and longevity requirements of edge computing can be improved by finding suitable edge cluster types to reflect the characteristics of the applications. To support effective resource usage and fast real-time response for edge computing, the proposed simulator identifies a suitable number of things in forming the edge cluster.
The topic is important and contributions in the area could be useful for numerous applications as the IoT has grown significantly and edge computing is one important component in the IoT ecosystem.
Comments:
- Overall, the structure and organization of the article are good and well-written.
- The exact type of Raspberry Pi the simulator trying to simulate should be clearly stated in the Introduction section not just in section 5.
- The contributions section is missing.
- A good overview of related works.
- Line 62: “an open-source development board called Raspberry Pi” This statement is not accurate. Raspberry Pi is not an open-source board, as not all hardware has full datasheets, but the OS is. For example, schematics are available but the PCB design is not published.
- Line 350: The official OS is now Raspberry Pi OS, not Raspbian.
- Line 505: “the similarity between the RaSim simulator and the actual Raspberry 504 Pi was confirmed through three factors: CPU usage, bandwidth, and battery usage.” Why these three metrics? Justify!
- Table 4, 5, 6: Consider replacing “Index” and “Explanation” with “Parameter” and “Description”, respectively. Also need to put the unit for each parameter.
- All the parameters and the numbers selected need to be justified. For example, why only 5,000 data considered for the ML experiment? 5,000 is considered a small number in ML.
- It would be interesting to see how the simulator compared with Raspberry Pi 4.
- It would be nice if the simulator can be put as open-source and share publicly.

Round 2
Reviewer 2 Report
The authors have addressed some of the previous concerns and improve the manuscript. However, some points still need to be addressed before possible publication:
Line 520: “Service providers consider computational capabilities (CPU), battery usage, and bandwidth of things to provide IoT services.” Please add references to support your statement.
Line 767: “The number of learning data used was 5,000, and each cluster was learned by distributing the data.” Why 5,000? Still need to justify this selection. Please add references to support your statement.
Author Response
Please see the attachment.

This manuscript is a resubmission of an earlier submission. The following is a list of the peer review reports and author responses from that submission.
Round 1
Reviewer 1 Report
The topic is interesting and falls within the scope of the journal.
The description of the simulator is missing several important pieces of information:
- the proposed simulator is not framed in the classification proposed by the authors themselves (full-stack simulator, Big Data processing simulator, network simulator). Is it a full-stack simulator? It appears not to cover adequately the processing tasks, while it looks focussed on the communications aspects. Hence it appears to fall short of being a full-stack simulator
- the estimation of the load due to processing tasks is not contemplated; how does the simulator compute how much processing power is required? Is the simulator capable of replicating all the processing tasks that may be required of the edge devices?
The analysis of the simulators performance is note conducted systematically. There is not a set of use cases. The analysis of processing requirements is limited to sorting, which is just one of the many processing tasks that may be required. In addition, it is not clear whether all the devices are assumed to carry out the same task, or whether the simulator is capable of considering tasks that differ among the several edge devices.
The interactivity claimed in the title appears to be completely missing. How is the simulator used to optimise the allocation of edge devices?
My suggestions are to
- provide a deeper description of the simulator, considering both the processing and the communication tasks
- describe its actual use to optimise the architecture of devices (which is what the authors claim)
- carry out its evaluation (or its demonstration) by defining a precise set of use cases, where multiple processing tasks are considered
Author Response
We are indebted to the reviewers for their valuable comments which were extremely helpful in revising our manuscript. We have given due considerations to each and every comment in preparing the revised manuscript.
For english language editing, we had proofreading process from professional english editing service(Editage).

Reviewer 2 Report
A large part of the related work section can be moved to the introduction section, keeping only the simulators' articles in the related work section.
It is necessary to discuss other proposals as IoTSim-Edge, YAFS, and others based on iFogSim. A shortlist can be found here: http://dataminer.me/2020/02/29/awesome-edge-computing/
The figure of simulator architecture needs to be improved, detailing the modules and connections between them.
Table I need to be better diagrammed. The same for Table II. I suggest the authors use the LaTeX template.
In section 4.4, it is not clear how much each field's size on the message structure. This kind of information is necessary to help the readers understand the overhead and other results obtained through the simulator.
In Figure 7, it is necessary to present clearly why the CPU consumption is (maybe equivalent, but) higher in RaSim. Is it would be because of the simulator overhead?
The proportion of average consumption in Figs. 8 and 9 appear to be similar, considering the real and simulated devices. Thus, is it possible to calibrate the simulator to correspond to the real device consumption?
The graphic figures need to be improved in quality. I suggest plotting using matplotlib or gnuplot.
Author Response

(The authors gave the same response as above.)

Reviewer 3 Report
The article describes an experiment based on a CPU, battery and bandwidth usage for virtual things that are created in a simulator, based on Raspbbery PI.
Figures:
- Figures 1 - 6, 11 - Is your creation? or it is from another source? If yes, please, cite it properly.
- Figure 7 - How did you compute the selection sort and bubble sort values? How did you obtain those percentage values? using the normalized equation from (1)? I am not sure if that percentage can be computed using (1). Bubble sort and selection sort are just sorting algorithms, so, the values that you are sorting how they were obtained. Consider improving these sections by explaining exactly how the values were obtained and give such examples of values. Maybe a table structure with those values will be good.
Tables
- Table 1 - recommend to be added on a single a page and not split it on two pages.
Author Response

(The authors gave the same response as above.)

Round 2
Reviewer 1 Report
The authors have complied successfully with just some of the comments raised about the first version of the paper.
The description of the single modules of the simulator has improved significantly.
However, the following issues still remain to be dealt with:
- The use-case employed in the simulation experiment is not described properly. A brief mention is given in the Conclusions (too late), while its description should be given in Section 5.2. It still appears in the Conclusions that just the use of the sorting function is contemplated, among the many processing tasks that edge devices may perform. The detailed description of the use case and a warning about the limitations due to using a single use-case based on sorting alone must be provided. Simulators should be evaluated against several use-cases. See, for example, the following paper: Sonmez, C., Ozgovde, A., & Ersoy, C. (2018). Edgecloudsim: An environment for performance evaluation of edge computing systems. Transactions on Emerging Telecommunications Technologies, 29(11), e3493. If your paper considers sorting tasks alone, that must be said explicitly and justified.
- There is still no explicit link between the simulator's experiment and its use in optimisation, as mentioned in the abstract. The authors do not define what "optimal" is for them, and in their response to comments, they just refer to the metrics they evaluate (number of rounds to death, number of transmission per thing, etc.). Well, please define what is optimal with respect to these metrics and what would the optimal configuration be after considering ALL the metrics.
In addition, please state explicitly in the Introduction what your claims are for your simulator with respect to the literature: which are the original features of your simulator with respect to what has been proposed in the literature?
Finally, please thoroughly check the paper for English. The paper contains even grammar mistakes like that at line 213 "When a thing not includes a sensor" --> " When a thing does not include a sensor".